# Melatonin in Neurodevelopmental Disorders: A Critical Literature Review

**DOI:** 10.3390/antiox12112017

**Published:** 2023-11-20

**Authors:** Cyrille Feybesse, Sylvie Chokron, Sylvie Tordjman

**Affiliations:** 1Pôle Hospitalo-Universitaire de Psychiatrie de l’Enfant et de l’Adolescent (PHUPEA), Centre Hospitalier Guillaume Regnier, 154 rue de Châtillon, 35000 Rennes, France; 2Integrative Neuroscience and Cognition Center (INCC), CNRS UMR 8002, Université Paris Cité, 45 rue des Saints-Pères, 75006 Paris, France; sylvie.chokron@cnrs.fr; 3Faculté de Médecine, Université de Rennes, 2 Avenue du Professeur Léon Bernard, 35000 Rennes, France

**Keywords:** neurodevelopmental disorders, brain development, melatonin, circadian rhythms, sleep–wake rhythms, sleep disturbance, autism, schizophrenia

## Abstract

The article presents a review of the relationships between melatonin and neurodevelopmental disorders. First, the antioxidant properties of melatonin and its physiological effects are considered to understand better the role of melatonin in typical and atypical neurodevelopment. Then, several neurodevelopmental disorders occurring during infancy, such as autism spectrum disorder or neurogenetic disorders associated with autism (including Smith–Magenis syndrome, Angelman syndrome, Rett’s syndrome, Tuberous sclerosis, or Williams–Beuren syndrome) and neurodevelopmental disorders occurring later in adulthood like bipolar disorder and schizophrenia, are discussed with regard to impaired melatonin production and circadian rhythms, in particular, sleep–wake rhythms. This article addresses the issue of overlapping symptoms that are commonly observed within these different mental conditions and debates the role of abnormal melatonin production and altered circadian rhythms in the pathophysiology and behavioral expression of these neurodevelopmental disorders.

## 1. Introduction

Melatonin (5-Methoxy-N-acetyltryptamine, N-acetyl-5-methoxytryptamine, NSC-113928) is a neurohormone synthesized in the pineal gland, and its secretion is enhanced by darkness and inhibited by light (daylight or artificial light). Indeed, Melatonin is often labeled as being the «darkness hormone» for having its peak secretion during the late evening hours while its production is suppressed by light [1,2]. Melatonin is a pleiotropic neuroendocrine molecule released in the brain at night and plays a crucial role in the synchronization of circadian rhythms, including sleep–wake rhythms, and controlling seasonal rhythms, including reproduction [3]. Melatonin is considered an endogenous synchronizer located in the suprachiasmatic nuclei (SCN) of the hypothalamus, stabilizing bodily rhythms and as a chronobiotic molecule that reinforces variations or adjusts the time period of the central biological clock [4]. It derives from serotonin to form first N-acetylserotonin (NAS) by acetylation through the enzyme arylalkylamine-N-acetyltransferase (AANAT, EC: 2.3.1.87) and then forms melatonin by methylation through the enzyme acetylserotonin-O-methyltransferase (ASMT, EC: 2.1.1.4). Serotonin also contributes to the development of the brain before acting as a neurotransmitter on mature brain; in particular, serotonin has a role on dendritic development and branching in the hippocampus and cortex [5]. Melatonin synthesis is regulated periodically in the SCN. This clock determines the circadian rhythm of melatonin secretion. Melatonin is a powerful natural antioxidant providing beneficial and protective effects against oxidative stress [6]; melatonin is found in mitochondria at concentrations higher than the ones in the blood, suggesting that melatonin could be viewed as a mitochondrial antioxidant [7]. A specific section on the antioxidant properties of melatonin is developed in this article. In addition, melatonin receptors can be found in the regions associated with the master circadian clock [8]. Melatonin has an important role in the circadian cycle, which is the temporal organization of physiological, cellular, neural, biochemical, and behavioral processes. It helps the body to anticipate the different phases of the day in a proactive way [9]. Melatonin affects various temporal processes via mainly high G protein-coupled melatonin receptors 1 and 2 (MT1, MT1) widely distributed across brain and peripherical tissues (for a review on melatonin receptors, see Tordjman et al. [10]). Many melatonin receptors are found in neuroendocrine and acoustic-vocal integration areas [11]. Animal models with diurnal birds or breeding green treefrogs changed vocal behaviors through melatonin interactions with receptors influencing both inhibitory and excitatory signaling; melatonin mediates the regulation of neural excitability in vocal-motor circuits. Scholars are now aware of the role melatonin can have on social communication and its development [12]. In the same line, as discussed below, melatonin is important for child development even before birth. 

## 2. Melatonin and Typical Neurodevelopment

Since the beginning of the normal pregnancy process, melatonin secretion is involved with the oocyte quality and the parturition course. The night-time concentrations start increasing after 24 weeks of gestation and reach significantly higher levels by 32 weeks. Melatonin receptors are widespread in the embryo and fetus since early stages. There is solid evidence that this neurohormone has a role in fetal neuroprotection, as normal sleep patterns are involved in human neurodevelopment even at this stage. It is noteworthy that melatonin is the regulating factor of the fetal sleep patterns development set in late pregnancy. Voiculescu et al. [13] found strong evidence that melatonin has a positive effect on the outcome of compromised pregnancies. Melatonin concentrations progressively increase in maternal blood during pregnancy, reaching their maximum at term. Researchers have also found melatonin in amniotic fluid [14]. In addition, chronic disruption leads to reproductive dysfunction and appears to be an important factor in the development of offspring diseases in adulthood (this relates to the concept of fetal programming). Melatonin decreases in conditions associated with serious outcomes for the fetus and appears to be involved in preeclampsia and intrauterine growth restriction [15]. Animal models of fetal growth restriction in newborn lambs showed that maternal administration of melatonin reduced fetal hypoxia, improved neurodevelopment, and decreased brain injury and oxidative stress [16]. Taken together, these studies suggest that the effects of melatonin on the development of human fetuses appear to be not limited to circadian rhythmicity. 

After birth, melatonin is secreted into the general circulation and in the cerebrospinal fluid, allowing this neurohormone to circulate throughout the body and the brain [17]. In the brain, extra-pineal melatonin behaves similarly to neurotrophic molecules [18,19] and is capable of modulating cell survival, proliferation, and differentiation by signaling pathways that can be triggered in response to stimulation of membrane and intracellular receptors. Thus, melatonin plays a crucial role in brain neuroplasticity and neurodevelopment via neurotrophic factors, promoting its growth and survival [20,21]. 

In addition, natural maternal melatonin is also a powerful free radical scavenger and an antioxidant protecting the baby and fetus within the maternal-placental-fetal unit [22]. Indeed, maternal melatonin deprivation during gestation or lactation has been shown to delay the infant’s physical maturation and neurobehavioral development. Melatonin found in the gastrointestinal tract of newborns is of maternal origin knowing that melatonin easily penetrates the placenta during the preterm period; melatonin is secreted into the mother’s milk after birth and may be involved in the production of meconium [2,23,24,25]. Melatonin also follows a circadian rhythm in human breast milk, and studies report undetectable levels of melatonin during the day and high levels at night [22,26]. Several studies reporting that melatonin rhythms set around 3 months of age in typical development allow us to understand better why infants begin at this period of life to have more regular sleep–wake cycles combined with regular night-time sleep lasting 6 to 8 h [23,27]. Similarly, the infant’s circadian cortisol rhythm is only set up at around 3 months of age [28]. The development of circadian rhythms occurs fully between 49-and 52-weeks post-conception and corresponds to the developmental phase when increased periods of deep sleep during the night are consolidated as infants have fewer nocturnal awakenings [23]. Without maternal melatonin, infants establish circadian rhythms mainly by neurological maturation [29]. Infants born prematurely or facing circumstances related to normal intrauterine development show a significant delay in pineal rhythmicity. In the life span, the highest melatonin levels are found in children younger than 4 years [30]. Infants benefit from increased cell proliferation differentiation and survival rates of novel neurons in the hippocampus when melatonin is administrated after birth [31]. It also plays a role in terms of having effects on excitation/inhibition balance by changes in neurotransmitter levels [32]. The balance between excitation and inhibition in synaptic inputs of neural circuits must be tight to avoid the pathogenesis of neurodevelopmental disorders [33]. Melatonin levels decline progressively with age, although circadian rhythms tend to be highly consistent day to day at any age [30]. Biological aging is a natural process leading to the disruption of circadian rhythms; aging is associated with the dampening of circadian gene expression, as aging is associated with an increase in oxidative stress [34]. 

The antioxidant properties of melatonin and its physiological effects have first to be considered in the next section to understand better the role of melatonin in typical and atypical neurodevelopment, as discussed in the following sections.

## 3. Antioxidant Properties and Physiological Effects

The antioxidant action of melatonin involved in the cardiovascular, immune, gastrointestinal, oncostatic, and brain-protective effects of melatonin is presented in this section. The protective effects of melatonin are summarized in Figure 1 and are developed below. 

Melatonin regulates blood pressure and autonomic cardiovascular and immune function, in addition to other physiological processes such as free radical detoxification and antioxidant effects via MT3 receptors, protecting the brain from oxidative stress [35,36,37,38,39,40,41,42,43].

Also, the antioxidant action of melatonin protects the gastrointestinal tract from ulcerations by (1) reducing hydrochloric acid secretion and oxidative effects of bile acids on the intestinal epithelium; (2) increasing microcirculation and bicarbonate secretion from duodenal mucosa via MT2 receptors (this alkaline secretion is an important mechanism for duodenal protection against gastric acid); and (3) fostering epithelial regeneration [4,44].

Melatonin has direct immuno-enhancement effects in both humans and animals, which is relevant to its function in immunological regulation [45,46]. The production of cytokines, and more precisely, certain interleukins (IL-2, IL-6, and IL-12), is selectively stimulated by melatonin [47]. Melatonin also improves T helper immune responses [46,48]. Additionally, the antioxidant properties of melatonin contribute to its immunostimulatory effects [47] and act indirectly by lowering the production of nitric oxide, which helps to reduce the inflammatory response [49].

Furthermore, given that oxidative stress is implicated in the origin, promotion, and development of carcinogenesis, the oncostatic protective effects of melatonin have been documented and linked to its anti-oxidative action [50,51].

Regarding brain protection, there is growing experimental evidence showing therapeutic benefits of melatonin for prematurity as well as for neurodegenerative conditions like Alzheimer’s disease, Parkinson’s disease, Huntington’s disease, and amyotrophic lateral sclerosis (for a review, see Polimeni et al. [52]). To determine the precise therapeutic concentrations required according to the specific disease, age of individuals, and brain lesion, as well as to examine the short- and long-term effects of melatonin on physiological, functional, and cognitive outcomes, further studies and clinical trials are requested in preterm neonates as well as aging adults. Finally, in addition to its therapeutic benefits for sleep problems, melatonin is of major interest regarding its antioxidant action, increasing brain protection against oxidative stress and inflammation, in general for atypical development, and in particular for neurodevelopmental disorders.

## 4. Melatonin and Atypical Neurodevelopment

The establishment and maintenance of several circadian rhythms, such as the one involved in the secretion of melatonin and sleep–wake rhythms, depend upon the interaction of light perceived by the retina and the suprachiasmatic nucleus [23]. Some stressful and traumatic situations experienced by pregnant women decrease maternal melatonin production, and this can have an impact on the internal rhythms and post-natal development of the fetus [24]. Brain anomalies such as the reduction of the hippocampus volume can occur during early childhood or before birth due to the impact of stress in this brain region [25]. In the same way, children in their primary infancy who are coping with stressful situations or traumas like insecure attachment, separation from the mother, or abuse can consequently develop brain anomalies [25]. These traumas have long-term effects on cognitive functioning [22]. Among children diagnosed with developmental disabilities, many of them (the frequencies range from 25% to 85%) also show sleep problems [26].

Tauman et al. [53] show relationships between low melatonin production in the first weeks of life and impaired psychomotor development by measuring nocturnal urinary excretion of 6-sulfatoxymelatonin. Melatonin production is related to brain functioning and has effects on neurological development, given its impact on the onset of circadian rhythms, as shown by a study on REM sleep (i.e., the rapid eye movement during sleep) [54]. The relationships between low 6-sulfatoxymelatonin excretion and developmental delay appear very early, including reports at 16 weeks of age [33]. 

## 5. Melatonin and Neurodevelopmental Disorders

Neurodevelopmental disorders encompass intellectual disability, autism spectrum disorder (ASD), or neurogenetic disorders associated with ASD, as well as schizophrenia or bipolar disorder occurring later in life, as these two last conditions are more and more considered as neurodevelopmental disorders [55]. These different disorders can be seen lying on a neurodevelopmental continuum and having considerable comorbidity, as observed in patients showing an overall deficit of melatonin production [56]. A high prevalence of altered circadian rhythms, including sleep–wake rhythms, was observed in individuals with these neurodevelopmental disorders [57], strengthening the interest to focus on melatonin metabolism in these disorders given the role of melatonin in sleep–wake rhythms, synchronization of circadian rhythms, and neural development.

### 5.1. Relationships between Melatonin and Neurodevelopmental Disorders in Infancy

Melatonin production and circadian rhythms have been consistently associated with mental disorders that occur in primary infancy. Early neurodevelopmental disorders, such as autism spectrum disorder (ASD), have been associated with a dysregulation of circadian cycles, especially the circadian cycle of melatonin production. It is noteworthy that alterations in circadian sleep–wake rhythms are frequently observed in these neurodevelopmental disorders with abnormalities in melatonin secretion. 

ASD is a behavioral syndrome with altered sensory motor development and sleep– wake rhythms [58,59,60]. Key behavioral features of ASD are characterized by impairments in social communication and restricted interests with repetitive patterns of behaviors [61,62,63]. This condition is often associated with common comorbidities such as intellectual disability, epilepsy, and severe sleep disorders [64]. Family home movies of infants who were subsequently diagnosed with ASD showed motor and emotional asynchrony between infants before 12 months of age and their parents [65,66]. These early signs are not specific to autism but offer indicators of atypical development, which become more evident in the second year of life [65]. In later stages of development, several signs were also reported, such as abnormal eye contact and other social communication impairments in learning through imitation (people’s faces, gestures, or vocal signals), social reciprocity, joint attention, and orienting to name or body language [67,68].

Melatonin is a common pharmacologic treatment used to deal with sleep disturbance due to circadian phase delay. The melatonin treatment provides a significant decrease in sleep latency and night awakenings and an increase in sleep quality and sleep efficiency (for a review, see Tordjman et al. [58,68]). However, Moon et al. [69] indicated that evidence of the therapeutic benefits of melatonin on psychiatric disorders is robust only in autism, attention deficit hyperactivity disorder (ADHD), and neurocognitive disorders. Sleep disturbances, such as falling asleep or having night awakenings, are relatively common among children with ASD, and their prevalence is higher when compared with children with typical development. This contributes to a variety of disturbances in their daily lives, such as behavioral problems, self-injurious behaviors, and other-injurious behaviors, and emotional problems like depression or anxiety. Finally, sleep deprivation among children with ASD also has negative ecological consequences since it affects parents’ or caregivers’ overall mental health. These difficulties can be related to odd bedtime routines and bedtime resistance [70]. 

Several studies reported that individuals with ASD showed lower melatonin levels in urine, plasma, and pineal gland than control groups [64]. Furthermore, several studies provided evidence of relationships between melatonin deficit and social communication impairments that are prevalent in neurodevelopmental disorders. In ASD, a lack of melatonin production was associated with language impairments [71,72]. The Tordjman et al. studies showed that abnormally low nocturnal melatonin excretion is significantly associated with severe autistic social communication impairments, especially verbal communication and social imitative play impairments in children and adolescents with ASD [73,74]. Moreover, this deficit in melatonin may be involved in ASD development through desynchronized, disrupted, and abnormal circadian rhythms but also through several physiological pathways, including a lack of antioxidant protective effects (as seen in Section 3 on antioxidant properties and physiological effects, melatonin protects the brain from oxidative stress and its antioxidant action decreases the production of nitric oxide which helps in turn to decrease the inflammatory response), and impairments in neurotransmission, synaptic plasticity and metabolic pathways [75]. It is noteworthy that nitro-oxidative stress, immune-inflammatory, neurotransmission, synaptic plasticity, and metabolic pathways are also under the control of the circadian clock [75]. In addition, researchers found metabolic disorders and neurochemical imbalances in the melatonin/serotonin system of children with ASD or Down syndrome [76]. It was suggested that the autistic, well-replicated hyperserotonemia [73] could cause a loss of serotonin terminals [77], possibly involved in autistic behaviors, given that serotonin production was associated with poor social interaction, emotional detachment, and aggression towards others [78]. Concerning Down syndrome, associations were observed with a serotonin deficit in the postmortem brain, cerebrospinal fluid, and blood [5].

Furthermore, altered melatonin circadian rhythms and impaired melatonin secretion are also reported in several neurogenetic disorders associated with autism, such as Smith–Magenis syndrome, Angelman syndrome, Rett’s syndrome, Tuberous sclerosis, or Williams–Beuren syndrome. A summary concerning the melatonin abnormalities found in these neurogenetic disorders associated with autism is presented in Table 1 and includes particular sleep problems, melatonin impairments, and the response to melatonin therapy observed in these developmental neurogenetic disorders.

### 5.2. Association of Melatonin with Mental Disorders Emerging in Early Adulthood 

Neuroscientists are trying to understand schizophrenia through a new approach involving neurodevelopmental maturation. Schizophrenia onset usually occurs in adolescence or early adulthood but might be related to vulnerability in infancy. Studies in individuals with a first episode of schizophrenia reported a decrease in grey matter volume for most of the examined brain regions and cerebellar area [133]. This reduced volume of grey matter can get even smaller and extend to other surrounding regions in chronic cases [134]. Individuals with schizophrenia suffer from changes in brain microstructure, physiology, and connectivity of widely acting neurotransmitter systems, resulting in affective, cognitive, and psychotic symptoms. Schizophrenia leads to major impairments regarding the more complex cognitive performances, the so-called higher-order cognitive functions like, for example, verbal episodic memory or executive functioning [135]. Individuals with schizophrenia show errors in integrative information processing that are hypothesized to result in mis-connectivity or dysconnectivity, leading to a dysfunction of multiple brain circuits [134]. Melatonin is viewed as an important biological marker of the circadian cycle and as a psychiatric therapeutic agent [136]. Individuals with schizophrenia show lower levels of nocturnal melatonin secretion compared to a healthy control group, poor sleep efficiency, and disrupted circadian rhythms [137,138]. The Galván-Arrieta et al. study on olfactory neuronal precursors in schizophrenia and typical development suggested that a deficit in melatonin may lead to impaired neurodevelopment in schizophrenia [139]. Brain autopsies revealed abnormal elevated HIOMT activity in schizophrenia due to abnormally low activity of an enzyme prior to HIOMT involved in the biosynthesis of melatonin [140]. The decreased endogenous secretion of melatonin can persist even if there are improvements in sleep and positive effects with psychotic agents [141]. Finally, the Beckmann et al. study [142] showed no abnormal melatonin concentrations in the cerebrospinal fluid of individuals with schizophrenia compared to healthy controls. However, the authors conclude that other possibilities, such as changes in biological rhythms related to variations in melatonin activity and its influence on other neuroendocrine functions, may have a role in the pathophysiology of schizophrenia.

Similarly, bipolar disorder can exhibit prodromal manifestation prior to illness onset, underlying eventual similarities of neurodevelopmental abnormalities possibly involved in the pathogenesis of bipolar disorder [143]. Bipolar disorder is mainly associated with a shift in mood, energy, and activity. Patients show sleep alteration, circadian cycle disturbances, emotional deregulation, cognitive impairment, and increased risk for comorbidities [144]. The psychopathology of bipolar disorder (BD) is associated with altered sleep/wake rhythms, thermoregulation, cortisol secretion, and melatonin secretion. BD patients show abnormal rhythmic activity that is more functionally impacted during inter-episode periods (the passage from a mania state to a depressive one) [145]. Circadian cycles are dysregulated regarding the sleep–wake rhythms, especially the number of hours of sleep [146]. Manic symptoms were associated with less robust circadian rhythms leading to a decreased need for sleep, but also with other symptoms such as thought disorder, increased rate and amount of speech, and increased motor activity and energy [147]. It is noteworthy that a manic phase with a decreased need for sleep is considered a critical marker of the appearance of a depressive phase [1]. During the depression phase, hypersomnia is prevalent in 100% of patients and is also followed by a delayed sleep onset with night-time awakenings [148,149]. It has been highlighted that bipolar disorder treatments could benefit from a better understanding of circadian cycles in bipolar disorder [1]. Therapeutic administration of agomelatine (a selective agonist of MT1/MT2 receptors and a selective antagonist of 5-HT2C/5-HT2B receptors) is of special interest given that agomelatine is involved in the resynchronization of interrupted circadian rhythms with therapeutic benefits on sleep patterns, resynchronizing circadian rhythms in individuals with depression (a few studies has also been published on ASD, ADHD, and anxiety) [150].

## 6. Transnosographic Approach on the Role of Melatonin in Neurodevelopmental Disorders 

Making an association between neurodevelopmental disorders and the pineal gland became more significant since the discovery of melatonin in the 1950s [148]. First, impairments in melatonin secretion have been associated with a significant decrease in sleep efficiency, notably in elderly individuals with continuity at different ages [6,10]. Second, melatonin seems to have a protective role in neurodevelopmental disorders with effects on early synaptic plasticity and neurotransmitter levels [33]. As indicated previously, circadian cycles have an impact on our bodies, shaping the timing and rhythms of various physiological and behavioral processes [1]. Children having an absence or alteration of circadian rhythms may have difficulties adapting to changes in their internal or external environment [151]. 

There is compelling evidence indicating that impairments in the endogenous circadian system, and especially in the sleep–wake rhythms (e.g., a delayed excretion of melatonin with a later onset of circadian rhythms), might precede the appearance of clinical symptoms for a variety of psychiatric disorders such as major depressive disorder, anxiety disorders or schizophrenia [152]. It suggests the existence of a biological dysfunction of rhythmicity and synchrony of rhythms in neurodevelopmental disorders. Reduced melatonin activity would then create a timing dysfunction of biological clocks with physiological and psychological disturbances leading to autistic social communication impairments. This timing dysfunction of biological clocks in early infancy could then lead to clinical psychopathological neurodevelopmental disorders later and even to psychotic or borderline psychotic states when the individual is emerging into adulthood. A disrupted sleep–wake cycle was associated as a major component for mood, anxiety, and psychotic disorders in adolescence [153]. Even if neurodevelopmental disorders display considerable genetic as well as environmental heterogeneity, severe mental conditions such as schizophrenia or bipolar conditions have increasing support from neuroscientific disciplines as originating from their origins from disturbed development of the nervous system based on the neurodevelopmental hypothesis [56,154,155]. New nosology conceptions should consider the genetic overlap between schizophrenia and psychopathologies associated with neurodevelopmental disorders that manifest in childhood [156]. Those conditions show some similarities in their phenotypes. All of them present significant cognitive impairment, tend to be more common among males, and are associated with developmental delay as well as neurological and motor abnormalities [155]. Symptoms and behaviors observed in autism are also found in adults with schizophrenia [157]. For example, autism and schizophrenia share intellectual disability, social communication impairment, or social withdrawal. One common factor that might be associated with these symptoms is a past or current deficiency of melatonin production.

There is a higher risk for ASD in children when the parents have mood disorders. The prevalence of ASD is higher in the offspring of parents with BD, then in the offspring of mothers with affective disorders, and finally in the offspring of parents with depressive disorders [158]. One possible explanation for these associations might be a possible shared genetic etiology between BD, depressive disorders, and ASD [159]. Reported family co-aggregation of BD and schizophrenia, as the common co-occurrence of ASD and BD, indicates a potential neurodevelopmental pathway [143]. The appearance of these neurodevelopmental disorders in families with ASD, BD, or schizophrenia was higher than in control families [160]. Further evidence to link the role of neurodevelopment in psychiatric disorders occurring later in life is the finding that neonatal lesions produce schizophrenia-like behaviors that emerge in post-adolescence [161].

In the Singh et al. meta-analysis, genetic data regarding rare coding variants in whole-exome sequences of 4133 schizophrenia cases and 9274 controls, de novo mutations in 1077 trios and copy number variants from 6882 cases and 11,255 controls, provided evidence that individuals with schizophrenia share rare damaging variants. Cases of schizophrenia with intellectual disability suggest that intellectual disability might be a dimension shared by neurodevelopmental disorders. These results support an overlap of genetic risk between schizophrenia and other neurodevelopmental disorders [162,163]. Several studies have concluded that schizophrenia is associated with cognitive impairments that probably result from a long-term neurodevelopmental evolution, even if it might be etiologically variable for each individual [135]. Risk factors associated with this disorder exert primarily their effects during developmental periods, leading to a detrimental maturation of the brain. The dysconnectivity in brain circuits found in schizophrenia is seen more and more as the result of abnormal brain development [134]. 

## 7. Conclusions

Psychiatric nosography suggests possible links between the pathogenesis of neurodevelopmental disorders and altered circadian rhythms. Altered circadian rhythms may participate in the development of these neurodevelopmental disorders but can also elicit and worsen psychopathological symptoms associated with this range of conditions. Melatonin production would consequently play a considerable role in sleep disorders and cognitive and social communication impairments. Neurodevelopmental disorders can greatly benefit from melatonin therapeutic administration and developmental behavior interventions that focus on rhythm synchronization [10,151]. Melatonin has been used with therapeutic efficacy for its resynchronizing effects to treat free-running rhythm disorder and delayed phase syndrome, among other circadian rhythm disorders [136]. Circadian disruption is a common prodrome for schizophrenia and bipolar disorder, as they are also for other mood disorders, but the nature of the relationships between melatonin, circadian rhythms, and psychopathology is still poorly understood today [164]. Further research is warranted to ascertain better the mechanisms underlying the effects of abnormal melatonin production and altered circadian rhythms on the pathogenesis and behavioral expression of neurodevelopmental disorders such as autism spectrum disorder or neurogenetic disorders associated with autism, schizophrenia, and bipolar disorder. Finally, given the neuroprotective and neurotrophic role of melatonin, it is a major issue to understand the relationships between the pathophysiology of melatonin metabolism and the development and severity of certain mental disorders discussed in the present paper.

## Figures and Tables

**Figure 1 antioxidants-12-02017-f001:**
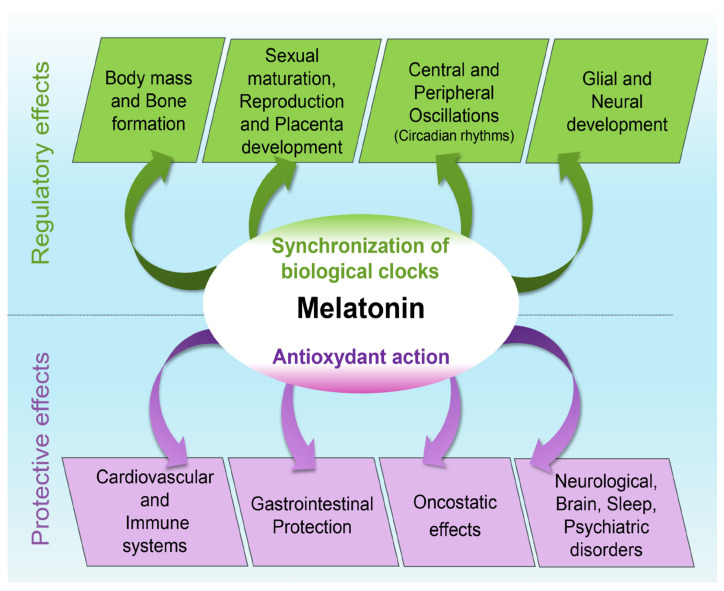
Antioxidant action and protective effects of melatonin.

**Table 1 antioxidants-12-02017-t001:** Melatonin abnormality and therapy in neurogenetic disorders associated with autism.

Neurogenetic Disorder	Neurogenetic Disorder Frequency, Estimated Rate (%) of Autism in the Disorder, and Estimated Rate (%) of the Disorder in Autism	Age of Diagnosis	Phenotype (Including Autistic Behaviors and Intellectual Functioning)	Sleep Problems	Melatonin Abnormality	Response to Melatonin Therapy
Smith-Magenis syndrome (SMS) Chromosome 17p11.2 microdeletion encompassing retinoic acid-induced 1 (RAI1) or a mutation in the *RAI1* gene [79,80,81,82]	- Frequency: 1 in 15,000–25,000 individuals [82,83] - Estimated rate of autism in SMS: 50–100 [84] - Estimated rate of SMS in autism: NA [84]	Many of the features of SMS are subtle in infancy and early childhood and become more recognizable with advancing age. Despite increased clinical awareness of SMS as well as improved cytogenetic technologies, many children are not definitively diagnosed until early childhood or even school age [85]	Facial dysmorphism, peripheral neuropathy, hypotonia, early feeding problems. Tantrums, self-injurious and stereotyped behaviors, sameness, developmental delay in vocalizations but possible social contact. Normal intellectual functioning to moderate intellectual disability [84,86,87]	Disrupted sleep patterns with shortened sleep cycles are characteristic of SMS and begin typically during the months after birth. Reports of excessive daytime sleepiness, increased sleep latency, frequent nocturnal and early morning awakenings due to an inverted circadian rhythm of melatonin [88]	Inverted circadian rhythm of melatonin secretion [88]	Melatonin therapy is used to regulate sleep problems. Combined with exogenous PRM (prolonged release melatonin), blockade of endogenous melatonin production during the day by the adrenergic antagonist acebutolol can improve impaired sleep and behaviors, and increase melatonin concentrations [89] Patients aged 3–18 years were given PRM (4 to 6 mg/day) as a single evening dose over a treatment duration of 6–72 months. Within 3 months, parents report improvement in sleep duration, sleep latency, number of midnight awakenings and sleep quality. No serious adverse events [88]
Angelman syndrome (AS) Maternal 15q11-q13 deletion, paternal uniparental disomy, mutations of *UBE3A* that encodes ubiquitin protein ligase (UBE3A) [90,91,92,93,94]	- Frequency: 1 in 12,000–20,000 individuals [95] - Estimated rate of autism in AS: 48–80 [84] - Estimated rate of AS in autism: 1 [84]	Developmental delays, between about 6 and 12 months of age, are usually the first signs, and seizures begin often between the age of 2 and 3 years old [96]	Facial dysmorphism, microcephaly, seizures (>1 year), ataxia and walking disturbance, Attention Deficit with Hyperactivity Disorder (ADHD), paroxysmal laughter, tantrums No language, stereotypies, sameness. Severe intellectual disability [84,96,97,98,99,100]	Severe sleep disturbances are common in Angelman syndrome, and are included in the diagnostic criteria [96]	The melatonin secretion profile of patients with Angelman syndrome is impaired, leading to a variety of sleep problems, most prominently in the areas of sleep-wake patterns and sleep duration [100]	Melatonin therapy significantly advanced sleep onset by 28 min, decreased sleep latency by 32 min, increased total sleep time by 56 min, and reduced the number of nights with awakenings from 3.1 to 1.6 nights per week [101]
Tuberous sclerosis complex (TSC), synonym: Bourneville disease (*TSC1*, 9q34) (*TSC2*, 16p13.3) Pathogenic variants in *TSC1* and *TSC2* genes: 31% and 69%, respectively [102]	- Frequency: 1 in 6000 individuals [103] - Estimated rate of autism in TSC: 25–60 [84] - Estimated rate of TSD in autism: 1–4 [84]	The average age at diagnosis of TSC is 7.5 years with 81% of patients diagnosed before the age of 10. Diagnosis may be difficult because symptoms are not present in all patients, and none are pathognomonic [104]	Autosomal dominant neurocutaneous disorder with ectodermal anomalies clinically diagnosed, renal lesions, seizures, learning disorder. Severe autistic syndrome. Variable intellectual disability [84,103,105,106,107]	Sleep problems are considered one of the most common behavioral manifestations in children with TSC [108]	Significant differences between TSC melatonin secretion profiles and control ones. Melatonin rhythm but not its amplitude was related to the total number of seizures [109]	Treatment improvement in sleep latency, total sleep time, and sleep fragmentation reported with melatonin at 5 mg dose [110]
Rett’s syndrome (RS) Mutation in the *MECP2* gene coding for the methyl CpG binding protein 2 and located at Xq28 [111,112]	- Frequency: 1 in 10,000–15,000 live female births [113] - Estimated rate of autism in RS: 61–100 [84,114,115] - Estimated rate of RS in autism: <1 in female [84]	Because of the apparent normal developmental course in early childhood, diagnosis may be delayed [116]	Developmental course: - Stagnation stage in girls (6–18 months); - Regression stage (12–36 months) with head growth deceleration, appearance of progressive motor symptoms (gait and truncal apraxia, ataxia, decreasing mobility) and respiratory symptoms (hyperventilation, breath holding, apnea); - Pseudo-stationary stage (2–10 years); - Late motor deterioration (>10 years). Autistic behaviors: stereotyped hand movements, absence of language, loss of social engagement. Severe intellectual Disability [84]	Sleep problems are common in Rett’s syndrome but there is some variation with age and mutation type [116]	Impaired secretion of melatonin [117]	Exogenous melatonin improved the sleep-wake cycle and sleep onset. The effect was maintained over 2 years without any adverse effects [117,118]
Williams Beuren syndrome (WBS) 7q11.23 deletion including 26 to 28 genes (typically *CLIP2*, *ELN*, *GTF2I*, *GTF2IRD1*, and *LIMK1*) [119,120]	- Frequency: 1 in 7500–10,000 [119] - Estimated rate of autism in WBS: <10 [84] - Estimated rate of WBS in autism: <1 [84]	The mean age at initial concerns is 0.98 year (Standard Deviation: 1.24), and the mean age at diagnosis may be delayed to 3.66 years (Standard Deviation: 4.13) [121]	Facial dysmorphism, short stature, heart and endocrine malformations, hypercalcemia, feeding problems, hyperacusis, visual spatial deficit, risk for attention deficit. Autistic syndrome but overfriendliness with social disinhibition and overtalkativeness. Mild to moderate intellectual disability [122,123,124,125,126,127]	Sleep disorders are common in individuals with Williams syndrome [128]	The WBS group had shallower drops in cortisol and less pronounced increase in melatonin at bedtime compared to the control group [129,130]	Melatonin was the most frequently reported medication taken for sleep problems in WBS, with 91% of parents reporting benefits for their child with WBS, and very few, if any, side effects [131,132]

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
