# Peer review of "Melatonin in Neurodevelopmental Disorders: A Critical Literature Review"

_antioxidants, 2023, doi:10.3390/antiox12112017_

Round 1

Reviewer 1 Report

Comments and Suggestions for Authors

The article under review, titled "Melatonin in Neurodevelopmental Disorders: A Critical Literature Review" ambitiously delves into the complex relationships between melatonin and various neurodevelopmental disorders. The authors attempt to connect melatonin's antioxidant properties and physiological effects to the development of both childhood and adult mental conditions. While the topic is undoubtedly intriguing, this review ultimately falls short of providing a convincing argument due to several critical flaws.

One of the most significant issues with this article is its lack of concrete evidence supporting the link between melatonin and neurodevelopmental disorders, missing the description of the complex pathway of melatonin synthesis, the mechanism of action that justifies elaborately the antioxidant potential of melatonin and its unique scavenging role, the extrapolation of the physiologic imbalances in its synthesis and MT receptors availability or sensitivity to pathological events, and nevertheless the complete description of the disease development based on melatonin misuse and the clinical trials that support this hypothesis. While the authors reference a wide range of conditions, from autism spectrum disorder to bipolar disorder and schizophrenia, they fail to present robust empirical data connecting these disorders to melatonin deficiencies or circadian rhythm disturbances. Instead, much of the discussion relies on speculative theories and theoretical connections rather than empirical research.

Furthermore, the article's attempt to address overlapping symptoms across various mental conditions feels underdeveloped and inconclusive. It raises the question of whether abnormal melatonin production and altered circadian rhythms genuinely play a central role in the pathophysiology of these disorders, but the authors fail to provide compelling evidence to support their claims. This leaves the reader with more questions than answers and undermines the article's overall impact. In addition, the only synthetic table, which was rather promising, seems incomplete, revealing data on a single pathology.

In conclusion, while the article's topic is undoubtedly intriguing and has the potential to contribute significantly to the general understanding of neurodevelopmental disorders, it ultimately fails to deliver a scientifically argumented framework. The lack of concrete evidence, underdeveloped arguments, and simplified writing style weight a lot in the relevance of such a review in the given scientific environment. As it stands, this article is not ready for publication and should be rejected until these critical issues are addressed and resolved.

Comments on the Quality of English Language

NA

Author Response

Dear Reviewer,

We have tried to structure better the manuscript, in particular by adding a section developing the antioxdant action of melatonin with a relevant Figure on the physiological and protective effects of melatonin.

In addition, we have  conducted an extended work on melatonin abnormalities and therapy in neurodevelopmental disorders  associated with autism and summarized this literature review in an updated Table.

We hope that these changes will improve the interest of the article.

Best regards,

The authors of the article

Reviewer 2 Report

Comments and Suggestions for Authors

The authors reviewed research data on a possible role of abnormal melatonin production and altered circadian rhythms in some neurodevelopmental disorders such as autism, schizophrenia and depression. In general, this is an interesting review on an important problem. However, it has some drawbacks which need to be eliminated.

Specific remarks:

1.     More basic information on various melatonin receptors and  receptor mechanisms of melatonin effects should be given. It is known that melatonin mainly exerts its effect through different pathways with G protein-coupled melatonin receptor 1 (MT1) and melatonin receptor 2 (MT2), and additionaly it acts as an antioxidant.

2.     How does melatonin interacts with  mechanistic pathways in ASD which include altered immune-inflammatory, nitro oxidative stress, neurotransmission and synaptic plasticity, and metabolic pathways (Abdul et al., 2021)?

3.     As indicated by Moon et al. in the paper entitled Role of Melatonin in the Management of Sleep and Circadian Disorders in the Context of Psychiatric Illness (Curr Psychiatry Rep. 2022 Nov;24(11):623-634) “a very limited number of well-designed trials with melatonin to correct sleep and circadian rhythms exist in psychiatric disorders, and the evidence for efficacy is robust only in autism, attention deficit hyperactivity disorder (ADHD), and neurocognitive disorders”. The present review need to be supplemented with some data on melatonin and ADHD.

4.     There is only one table with a rather limited information. It should be expanded  with other data. The manuscript would also benefit from a graphical abstract and hihlights.

5.     The authors wrote on page 7: “It has been highlighted that bipolar disorder treatments could benefit from a better understanding of circadian cycles in bipolar disorder [1].” Do antipsychotic and antidepressant drugs recover (normalize) physiological circadian cycles of melatonin concentration in psychiatric patients? Perhaps agomelatine should be mentioned as it is  a selective agonist of melatonin receptors MT1 and MT2, and a selective antagonist of 5-HT2C/5-HT2B receptors and is involved in the resynchronization of interrupted circadian rhythms. resynchronizing circadian rhythms in patients with autism, ADHD, anxiety, and depression (Savino et al., Brain Sci, 2023)

6.     Some sentences need to be corrected or restructured, e.g.

Page 6: Melatonin is viewed as a biological marker important regarding the circadian cycles and as a psychiatric therapeutic agent [139].

        Page 6: The Galván-Arrieta et al. study on olfactory neuronal precursors in schizophrenia and typical development suggested also that a deficit in melatonin may lead to an impaired neurodevelopment involved in the developmental process of schizophrenia [142].

       Page 7: Patients are strike by sleep alteration, circadian cycle disturbances, emotional deregulation, cognitive impairment, and increased risk for comorbidities [147].

         Page 7. There are compelling evidence indicating…

Even if neurodevelopmental disorders display a considerable genetic as well as environmental heterogeneity, severe mental conditions such as schizophrenia or bipolar conditions started to be considered with more and more support from neuroscientific disciplines as having their origins from a disturbed development of the nervous system based on the neurodevelopmental hypothesis [57,157,158].

Author Response

Reviewer's comment 1:

 More basic information on various melatonin receptors and  receptor mechanisms of melatonin effects should be given. It is known that melatonin mainly exerts its effect through different pathways with G protein-coupled melatonin receptor 1 (MT1) and melatonin receptor 2 (MT2), and additionaly it acts as an antioxidant.

Response: 

More information on melatonin receptors and their related mechanisms is available as indicated in the following sentence added in green color to the revised manuscript with the reference of an article written by several authors of the present article:

" Melatonin affects various temporal processes via mainly high G protein-coupled melatonin receptors 1 and 2 (MT1, MT1) widely distributed across brain and peripherical tissues (for a review on melatonin receptors, see Tordjman et al. [10]). "

Reviewer's comment 2:

How does melatonin interacts with  mechanistic pathways in ASD which include altered immune-inflammatory, nitro oxidative stress, neurotransmission and synaptic plasticity, and metabolic pathways (Abdul et al., 2021)

Response:

The following sentences have been added in green color in the revised manuscript  (the reference [76] is the Abdul et al., 2021 one):

" Moreover, this deficit in melatonin may be involved in ASD development through desynchronized, disrupted and abnormal circadian rhythms but also through several physiological pathways including a lack of antioxidant protective effects (as seen in the section 3 on antioxidant properties and physiological effects, melatonin protects the brain from oxidative stress and its antioxidant action decreases the production of nitric oxide which helps in turn to decrease the inflammatory response), and impairments in neurotransmission, synaptic plasticity and metabolic pathways [76]. It is noteworthy that nitro oxidative stress, immune-inflammatory, neurotransmission, synaptic plasticity, and metabolic pathways are also under the control of the circadian clock [76]. "

Reviewer's comment 3:

  As indicated by Moon et al. in the paper entitled Role of Melatonin in the Management of Sleep and Circadian Disorders in the Context of Psychiatric Illness (Curr Psychiatry Rep. 2022 Nov;24(11):623-634) “a very limited number of well-designed trials with melatonin to correct sleep and circadian rhythms exist in psychiatric disorders, and the evidence for efficacy is robust only in autism, attention deficit hyperactivity disorder (ADHD), and neurocognitive disorders”. The present review need to be supplemented with some data on melatonin and ADHD.

Response:

We have added the following sentence  in green color in the revised manuscript: 

" The melatonin treatment provides a significant decrease in sleep latency and night awakenings and increase in sleep quality and sleep efficiency (for a review, see Tordjman et al. [60,69]). However, Moon et al. [70] indicated that evidence of therapeutic benefits of melatonin on psychiatric disorders is robust only in autism, attention deficit hyperactivity disorder (ADHD), and neurocognitive disorders. "

However, we did not develop data on melatonin and ADHD because it is a complete extended topic and we have restrained the topic of the article to neurodevelopmental disorders occurring during infancy such as autism spectrum disorder or neurogenetic disorders associated with autism (including Smith-Magenis syndrome, Angelman syndrome, Rett’s syndrome, Tuberous sclerosis or Williams-Beuren syndrome) and neurodevelopmental disorders occurring later in adulthood like bipolar disorder and schizophrenia.

Reviewer's comment 4:

  There is only one table with a rather limited information. It should be expanded  with other data. The manuscript would also benefit from a graphical abstract and hihlights.

Response:

The Table has been expanded and a Figure has been added.

Reviewer's comment 5:

The authors wrote on page 7: “It has been highlighted that bipolar disorder treatments could benefit from a better understanding of circadian cycles in bipolar disorder [1].” Do antipsychotic and antidepressant drugs recover (normalize) physiological circadian cycles of melatonin concentration in psychiatric patients? Perhaps agomelatine should be mentioned as it is  a selective agonist of melatonin receptors MT1 and MT2, and a selective antagonist of 5-HT2C/5-HT2B receptors and is involved in the resynchronization of interrupted circadian rhythms. resynchronizing circadian rhythms in patients with autism, ADHD, anxiety, and depression (Savino et al., Brain Sci, 2023)

Response:

The following sentence has been added in green color in the revised manuscript (the reference [153] is the Savino et al., 2023 one):

" Therapeutic administration of agomelatine (a selective agonist of MT1/MT2 receptors and a selective antagonist of 5-HT2C/5-HT2B receptors) is of special interest given that agomelatine is involved in the resynchronization of interrupted circadian rhythms with therapeutic benefits on sleep patterns, resynchronizing circadian rhythms in individuals with depression (a few studies has been also published on ASD, ADHD, and anxiety) [153]. "

Reviewer's comment 6:

Some sentences need to be corrected or restructured, e.g.

Page 6: Melatonin is viewed as a biological marker important regarding the circadian cycles and as a psychiatric therapeutic agent [139].

        Page 6: The Galván-Arrieta et al. study on olfactory neuronal precursors in schizophrenia and typical development suggested also that a deficit in melatonin may lead to an impaired neurodevelopment involved in the developmental process of schizophrenia [142].

       Page 7: Patients are strike by sleep alteration, circadian cycle disturbances, emotional deregulation, cognitive impairment, and increased risk for comorbidities [147].

         Page 7. There are compelling evidence indicating…

Even if neurodevelopmental disorders display a considerable genetic as well as environmental heterogeneity, severe mental conditions such as schizophrenia or bipolar conditions started to be considered with more and more support from neuroscientific disciplines as having their origins from a disturbed development of the nervous system based on the neurodevelopmental hypothesis [57,157,158].

Response:

All the sentences indicated by the reviewer have been corrected and the changes are highlighted in green in the revised manuscript. In addition, an English native speaker went through the manuscript and made the necessary corrections.

Round 2

Reviewer 1 Report

Comments and Suggestions for Authors

The article was significantly improved, a lot of requested and solicited data, details, tables and figure were added in order to develop a complex work, with important impact on the highly specialised reading public.

Reviewer 2 Report

Comments and Suggestions for Authors

The authors have addressed the referee's comments and the manuscript has been sufficiently improved.